# Land Resources Evaluation for Damage Compensation to Indigenous Peoples in the Arctic (Case-Study of Anabar Region in Yakutia)

**Sergey I. Nosov [1], Boris E. Bondarev [2], Andrey A. Gladkov [3] and Violetta Gassiy [4,*]**

[1] Department of Project and Program Management, Plekhanov Russian University of Economics, Stremyannoy Pereulok, 36, Moscow 117997, Russia
[2] Department of Agro Engineering, Peoples' Friendship University of Russia, Miklukho-Maklaya st., 6, Moscow 117198, Russia
[3] Department of Soil Science, Geology and Landscaping, Russian State Agrarian University—Moscow Timiryazev Agricultural Academy, Timiryazevskaya st., 49, Moscow 127550, Russia
[4] Public administration department, Kuban State University, Stavropolskaya st., 149, Krasnodar 350040, Russia
* Correspondence: vgassiy@mail.ru; Tel.: +7918-438-11-28

**Abstract:** The compensation for losses caused to the indigenous peoples in Arctic Russia due to the industrial development of their traditional lands is an urgent question whose resolution requires development of new mechanisms and tools. The losses caused to indigenous traditional lands are part of the damage caused to the natural environment, their culture and livelihood. In the Russian Federation cultural impact assessment is a rather new tool aiming to protect indigenous peoples' rights to lands. In this paper the authors show the applied side of the cultural assessment that is used to improve the methodology of the calculation of losses adopted by ministry of regional development in Russia in 2009. This methodology is based on the resource disposition and evaluation of traditional lands. Accordingly, compensation payments are calculated as the sum of the losses in traditional economic activities such as: reindeer herding, hunting, fishing and gathering. Such compensation is considered by authors as the elements of a benefit-sharing system. In practice, this methodology has been tested at industrial projects on alluvial diamonds in Yakutia. In this paper we look at the Polovinnya project case-study which deals with indigenous peoples of Dolgans and Evenks and argues that such a justified, understandable methodology both for indigenous peoples and subsoil user could reduce to a minimum the conflict of interests.

**Keywords:** indigenous peoples; benefit sharing; biological and economic reserves; resource assessment; land evaluation; compensation payments; Arctic; Yakutia

---

## 1. Introduction

The Northern territories of Russia are mostly dependent on the extraction of mineral resources. Due to negative financial situation they are forced to look for approaches to the adaptation of their socio-economic systems. One of the main approaches includes the formation of a regional investment platform aimed at the implementation of "anchor" projects. It must subsequently have a multiplier effect in socio-economic terms. Among such positive results are new jobs, the attraction of highly qualified specialists to the region, tax revenues, infrastructure development. Of course, in a situation of unstable foreign economic situation, it is important to focus on projects that can act as a driver of socio-economic development of the territory.

The decision to create core zones in the Arctic will serve as a comprehensive development of the regional economy [1]. The main idea is to consider the territory as a single investment project,

including interrelated production, infrastructure and social systems. This approach provides business inclusion in the process of core zones development, whose interests are also widely represented in the Arctic zone. In 2016 the list of priority investment projects was adopted by the State Commission for Arctic development [2].

The presented structure demonstrates a significant predominance of the raw material factor in the specialization of investment projects (Figure 1). Unfortunately, such areas as energy (5.0%), which has huge innovative potential and social significance for the Arctic territories, is relatively small compared to the extraction of minerals. Energy projects are capital-intensive, and important for the environment and social sphere, but cannot provide a significant economic effect in extraterritorial terms. In contrast to these capital-intensive projects, investments in the environment and social sphere are insignificant, although the innovations in this area can significantly improve the quality of life of the Arctic population. Currently, human settlements depend on the Northern delivery system, energy supply which affects the environment.

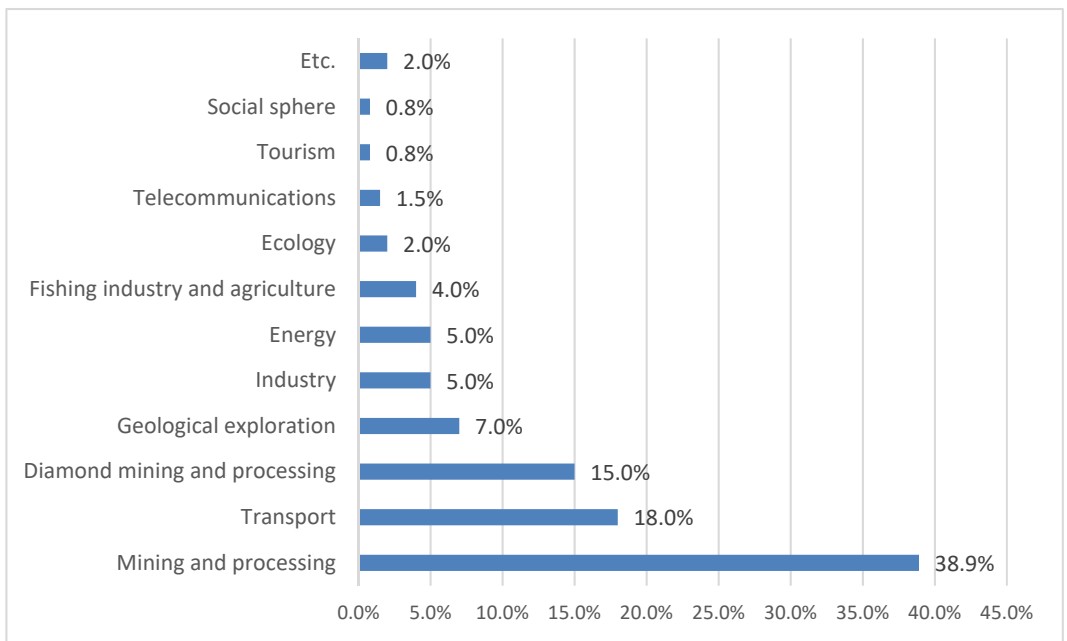

**Figure 1.** List of priority projects in the Russian Arctic [2].

"Anchor" projects are designed not only to lead to the structural changes in the economy of the region in which they are implemented, but also to affect the entire Arctic macro region. In fact, the core zones are created to provide the Northern sea route, transportation of transit cargo, as well as extracted minerals. In the structure of such system-forming investment projects there are the Arkhangelsk and Murmansk regions, Yamal-Nenets Autonomous area. The development of the Northern latitudinal route, the development of the Prirazlomnoye field, the start of the Yamal LNG-2 are the basic investments that determine the nature of Russia's strategic interests in the Arctic [3].

That means the industrial development will further move forward to the territories that were untouched in centuries to affect traditional lands and culture. It requires the development of partnership models between business and indigenous communities as well as the transparent system of losses calculations and impact assessment. If in Canada or US such systems have been successfully working and economic rights of indigenous peoples are clearly defended. In these countries there is a clear division of territories according to the principles of ownership, the status of indigenous communities and their rights to traditional lands. In Russia there is still the gap in the legislation of the tribal families' ownership of the traditional lands. At present, the issue of the conditions for the land provision to indigenous peoples for the maintenance of traditional lifestyles and management is regulated by the

legislative acts ambiguously and incompletely. It would seem that special legislation establishes the right of minority peoples to the free use of land in order to protect their traditional way of life, business and crafts. It is contrary to other legislation, where restrictions are provided. Thus, according to Article 10 of the Federal Law No. 101-FZ "On the turnover of agricultural land", land plots from agricultural land that are state or municipal property may be leased to indigenous communities. At the same time, they also cannot buy out the leased land in the property. It is the same situation with hunting. Today, the hunting grounds are put up for auction according to the federal legislation. If the indigenous hunter is without stable work today how he could buy out these lands? Fishing is the main form of survival and nutrition in the Arctic and other Northern regions. The transition of the fishery to auction also outraged the population. These are the conflicts where the law on the one hand protects indigenous peoples, and on the other the Forest Code infringes upon their right to traditional and ancestral lands. The indigenous peoples living in territories that have received the status of specially protected areas should not be restricted to the extraction of hunting resources for personal consumption. Today, northerners feel uncomfortable and cannot freely hunt in the tundra or fish, waiting for fines each time.

The traditional occupations of the indigenous peoples of the North—reindeer herding, fishing, hunting, are closely interrelated with the survival of these peoples in their native land. Here we can recall the Makivik Corporation and its highly developed system of business collaboration. In the Canadian province of Quebec, the Makivik Corporation ("The Makivik Corporation") is a significant economic force, an ethnic corporation whose investment interests are represented in areas such as oil and gas production, transportation, environmental protection, etc. Owning equity stakes in these areas over 30 years, the Corporation invested more than $ 100 million in social infrastructure and community support projects. In 2002, signed a trilateral agreement for 25 years on partnership in the field of socio-economic development of Nunavik (Nunavik) territory joined in 1999 in Canada—a place of compact residence for the indigenous people of the Inuit (Eskimos). The parties to the signing were the Government of Quebec, the regional authorities (The Kativik Regional Government) and the Makivik Corporation. According to this agreement, the main objects of joint investment are the mining industry, tourism, transport and social infrastructure, construction of hydroelectric power plants, environmental protection [4].

In Russia there is still a huge gap in impact assessment tools especially in the context of the traditional economy and cultural damage. This problem is connected with the absence of cultural assessment legislation which could be used in whole territory of Russia. Nowadays Yakutia is the one region where such norms have been adopted and indigenous communities could consider fair compensation for damage to traditional lands. They call this ethnological expertise—a comprehensive scientific research on social, economic and cultural impact of the investment project [3]. In the paper the authors describe the case-study of such impact assessment in very narrow terms. The biggest problem of such an assessment tool is the absence of clear indicators for non-material characteristics of indigenous culture. Thus, to do so the adopted methodology offers to calculate the possible losses of indigenous peoples through their traditional economic activities: reindeer herding, hunting, fishing and gathering. Traditional culture, customs and sacred places must be also considered and described. The idea is that calculation of losses allows definition of the compensation amount which could be used for the traditional culture preservation as well as efforts to fix social and economic problems. Obviously, four indicators mentioned above are connected with traditional lands as the main assets for indigenous livelihoods. That is why the land evaluation is considered as the basis of fair compensation calculation for benefit-sharing agreements. The authors want to concentrate your attention on the Russian specificity of cultural assessment and damage compensation in the Arctic to emphasize the need for its improvement and further discussion.

## 2. Materials and Methods

### 2.1. Losses Calculation and Compensation Payments as Discussion Points in Russian Science

In Russia the need of dialogue between the industrial companies and indigenous peoples is under active discussion. According to V.I. Shadrin, the vice-president of the Association of Indigenous peoples in Yakutia, the subsoil-users must be more inclusive to the indigenous communities' interactions saying that " ... the required conditions are the informing about the companies' intended activities and possible environmental impact, consulting and holding public hearings, as well as ethnological expertise" [5]. As the investment projects supposes the temporary withdrawal part of traditional land, the lost profit approach defines conceptual model of the calculation of losses and compensation. In Russia the problems of fair compensation to indigenous peoples are complicated by a lack of improvement of the legislation:

(1) Not all indigenous peoples in Russia are included in the list of indigenous groups officially registered in the country. It depends on some limits to the self-identification people as "indigenous" (population must be less than 50,000 etc.). But they could manage traditional economic activities and their livelihood could be affected by industrial projects of natural resources extraction as well. These issues can lead to contradictions and conflicts in the Arctic territories as the economic rights of such ethnic groups are not secured. As the option of the government decision-making could be to review the Federal Law of 30 April 1999 No. 82-FZ "On Guarantees of the Rights of the Indigenous peoples of the Russian Federation" on changing the terms of the registration procedure of the indigenous peoples" (Article 7) [6].

(2) The compensation mechanism is not well developed in Russia that also impacts the economic rights and welfare of the indigenous peoples. It is due to the methodology adopted in 2009 [6]. Its indicators are not considering non-material characteristics of the area as well as they are very complicated for calculation. But if the calculation methodology has been adopted, there is no clear mechanism on compensation payments. The legislation has the gap in who and to whom must pay or be paid, what kind of compensation could be (monetary or non-monetary). The business often uses this lack to avoid of additional costs as they considered compensatory payments. Such situation results in conflicts and negative consequences in the Arctic. The government should improve the methodology in a way of clear the recipients of the compensations [6]. The Federal Law No. 82-FZ of 30 April 1999 "On Guarantees of the Rights of the Indigenous Minorities of the Russian Federation" states that indigenous peoples have the right " ... to compensate for losses caused to them as a result of damage to the original habitat ... " [7]. This law is still not equipped with tools for its realization.

Nowadays in Russia there many scientists involved in losses calculation and compensation payments discussion. We can name Burtseva E.I., Kurakin V.I., Neustroeva A.B., Novikova N.I., Potravny I. M., Samsonov I.V., Kharyuchi S.N., Yanina D.V. and etc.

Novikova N.I., a leading researcher at the Institute of Ethnology and Anthropology of the Russian Academy of Sciences, believes that the amount of losses and compensation payments should be determined during ethnological expertise which must be carried out exclusively by ethnologists [8]. We cannot agree with such a position as the comprehensive study on cultural assessment requires the qualifications in geobotanical mapping, land evaluation, economic and data analysis as well as archeology, biology and computing modeling.

The problems of interaction between the indigenous peoples and mining companies in Yakutia are considered in the papers of Irina Samsonova. The scientist notes the need of legislation improvement for transparent system development between the indigenous peoples and business [9]. Kharyuchi justified the need to secure indigenous rights during Arctic development [10].

Ivan Potravny insists that the compensation mechanisms must be adopted at the federal level and it could have monetary and monetary nature [11]. Due to the Russian tax system on natural resources

extraction, revenue is payed to federal and regional budgets. Such a system misses the local and indigenous communities directly affected by the industrial projects. The compensation system could provide monetary inflows for a local economy and members of the indigenous communities. The indigenous communities could form the foundation for future generation to secure the times when the mine will be closed, to promote sustainable development. Also, the business could construct the social and transport infrastructure which are considered by Potravny as the non-monetary compensation. Burtseva notes that there is a strong need to create a clear, easy-to-understand algorithm for both industrialists and the indigenous communities to calculate losses [12]. The approved methodology causes a lot of complaints. The main ones are the complex calculation algorithm; inconsistency of the calculation methodology with the availability of the necessary source materials for surveys of traditional environmental management areas [13].

*2.2. Resources Assessment for the Losses' Calculation: the Way to Compensation for Damage to Traditional Lands*

The authors believe that calculation of losses must be done on the basis of resources assessment. According to Russian legislation there are 13 types of the traditional economic activities of the indigenous peoples. They include large variety of species and differ by their location:

1. Livestock, including nomadic (reindeer, horse, Yak, sheep).
2. Processing of livestock products, including the collection, harvesting and dressing of skins, wool, hair, ossified horns, hooves, antlers, bones, endocrine glands, meat, offal.
3. Dog breeding (breeding reindeer, sled and hunting dogs).
4. Breeding of animals, processing and sale of products of animal husbandry.
5. Beekeeping.
6. Fishing (including marine hunting) and the sale of aquatic biological resources.
7. Commercial hunting, processing and sale of hunting products.
8. Agriculture (gardening), as well as breeding and processing of valuable medicinal plants.
9. Harvesting of wood and non-wood forest resources for their own needs.
10. Gathering (harvesting, processing and sale of food forest resources, collection of medicinal plants).
11. Extraction and processing of common minerals for own needs.
12. Arts and crafts (blacksmithing and iron-making craft, manufacture of utensils, equipment, boats, sledges, other traditional means of transportation, musical instruments, birch bark products, stuffed commercial animals and birds, souvenirs from the fur of deer and commercial animals and birds, other materials, weaving from herbs and other plants, knitting nets, bone carving, wood carving, sewing national clothes and other crafts related to the processing of fur, leather, bone and other materials).
13. Construction of national traditional dwellings and other structures necessary for the implementation of traditional economic activities [14].

It should be noted that for most of the types listed above, there is no data. In this connection it is necessary to:

- create a modern up-to-date information and reference database on the productivity of reindeer pastures and land for the collection of wild plants, hunting and fishing grounds;
- update cartographic material: geobotanical maps, deer pasture maps, maps of types of hunting grounds. For example, in Yakutia the geobotanical maps were last updated in 1978;
- oblige the local authorities to organize data collection on traditional economy, as well as to ensure public access to this information [15].

Resources assessment allows for calculating the entire biological reserve of resources in the territory. In determining losses, only a part of it that is permissible to be withdrawn during traditional economic activities is taken into account. This part is called the economic reserve and its withdrawal

does not disturb the biological balance of the ecosystem. Losses are calculated on the entire area of the withdrawing land. For reindeer herding and hunting, the calculation of losses takes into account the stress impact on deer and hunting animals in the territory affected by the investment project. The object of the research is the territories of the traditional economic activities of the indigenous peoples, where industrial development is carried out by mining companies. In Russia the specificity of the indigenous peoples is reindeer herding as the main type of traditional activity compared to Canada or the US. In 2018 the livestock number reached 1.5 million and it still keeps traditional knowledge (language, nomad, equipment) [16].

In Russia there are five types of land resources used for reindeer herding [17]:

- interzonal (tundra–forest-tundra–northern taiga);
- tundra;
- taiga;
- intrazonal mountain-taiga;
- Far Eastern mountain-taiga seaside.

According to its nomadic character reindeer herding occupies millions of square miles of Arctic area in Russia. So, it faces with industrial development as no other traditional activity. The reindeer pastures affected by mining projects, change due to partial withdrawal, stress zones and pollution. In other traditional activities the radius of indigenous presence will be narrowed thus hunting grounds or fishing don't require to move on such long distance. The most important indicator of the productivity of reindeer pastures is the reindeer carrying capacity. It differs by seasons of breeding.

The resource assessment of the territories of traditional nature use is based on the calculation of income and costs of traditional economic activities. This method has been tested by authors for different investment projects on alluvial diamonds and gold extraction which has been realized in the Arctic in 2015–2019. Here we consider a case-study of the project on exploration and processing the alluvial diamonds on the Polovinnaya River. It is a tributary of the Anabar river in the North-West of Yakutia, flows into the Laptev Sea. The averaged geographic coordinates of the license area are: 72° 31' nl. and 114° 34' el. (Figure 2).

The total length of the assessed area along the Polovinnaya and its tributaries is 75.4 km. The square of the licensed (exclusion) area is 9160 ha.

During losses calculation, the side-effects of the industrial development must be considered. Due to "noise pollution", from the presence of a large number of people and equipment in this area, the animals experience discomfort and leave the area. Such an area of the land is called a stress zone. In case of Polovinnaya the outer boundary of the stress zone passes at a distance of 2.0 km from the border of the exclusion zone (the licensed area) along its entire length. The area of the stress zone is 21,049.9 ha. The total area of the site to which the negative impact is distributed is 30,209.9 ha. The researched deposit is located on the territory of traditional nature use, where the main type of traditional economic activity is reindeer herding, fishing, seasonal hunting and gathering.

To calculate the losses, the resource assessment was made for each type of traditional nature use. The data of economic stock of biological resources was used. An economic stock is a part of a biological one that can actually be taken (withdrawn) in a given territory without causing harm to the ecosystem. Resource assessment of the territories of traditional nature use is carried out in the following sequence:

- carrying out landscape and ecological zoning;
- geobotanical survey;
- resource assessment of territories.

The purpose of landscape-ecological zoning is the allocation and mapping of the contours of the biological resources accounting.

Landscape-ecological zoning is carried out on a topographic map at a scale of 1:100 000 with the involvement of colored aerospace photographs, applied regional maps: geological, geomorphological, soil, climatic, vegetation, landscape.

There are contours of accounting in the form of relief (meso-, micro -) and characteristics of vegetation as the main features that determine the eco-system.

At the same time, it is necessary to take into account the features of the hydraulic network, soil drainage and natural soil moisture, the power of the active layer of permafrost; weather and microclimatic conditions in zones, subzones, bands and belts (in mountainous areas); spots and strips of bare soil as a manifestation of seasonal activity of permafrost.

A geobotanical map is based on topographic maps using maps of landscape and ecological tours. All these methods were implicated in the researched area.

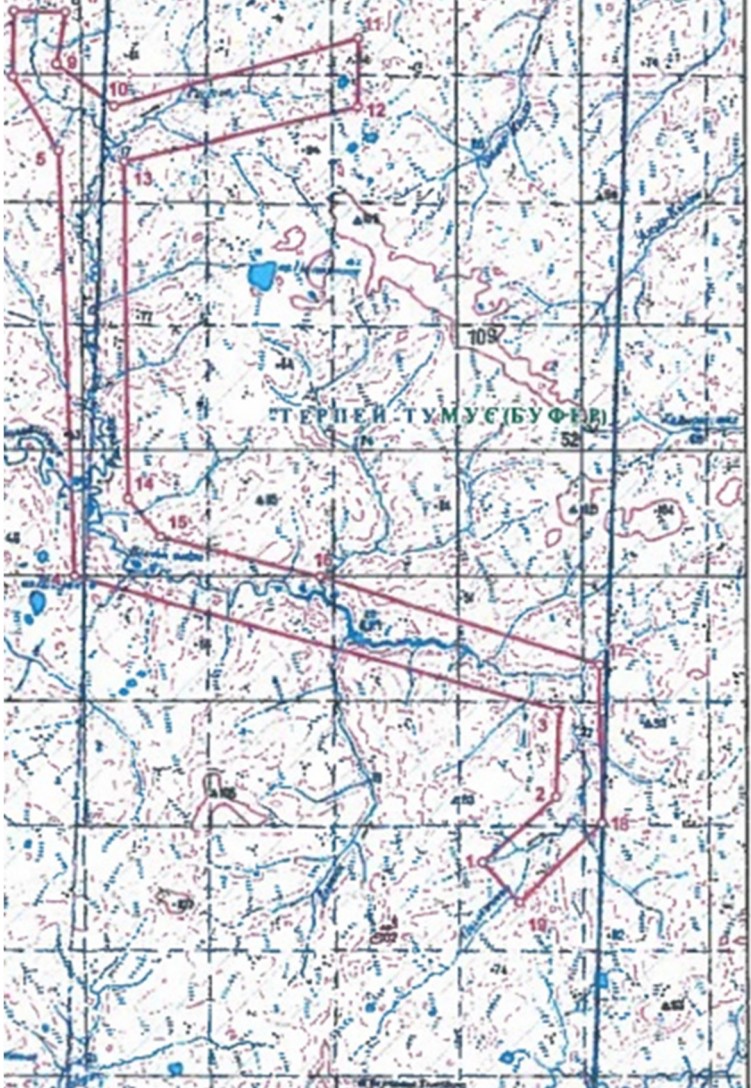

**Figure 2.** Map of the Polovinnaya licensed area (made by authors during cultural assessment)**.**

## 3. Results

Here we consider a case study of losses assessment for each type of traditional economic activity. The data was collected during Arctic expedition to the Anabar Dolgan-Even national region in Yakutia in 2017.

### 3.1. Calculation of Losses for Reindeer Herding

The losses of reindeer herding are calculated for each geobotanical contour of reindeer pastures. In fact, they are lost profit, which is determined on the basis of the annual net income lost.

The methodology supposes that due to withdrawal traditional lands and stress impact the indigenous peoples receive less profit.

The electronic map of the area was created on the basis of the raster image of the economic-geobotanical map. It includes different geobotanical contours divided by exclusion or stress zones. Each geobotanical contour is characterized by its productivity - reindeer carrying capacity, expressed in deer days per 1 ha. There will no reindeer herding on pastures withdrawn for industrial use. All lost income from these lands relates to losses. In the stress zone, there could be traditional economic activity. But due to the negative impact the productivity in such lands is reduced. According to expert data it decreases by a factor of two. The legislation on cultural assessment requires to use market prices of the traditional products in losses calculation. Information about the market value of reindeer husbandry is taken according to the statistics data, open publications and analytical studies.

In determining the gross value of the lost products of reindeer husbandry, not all the livestock is taken into account, but only its part that allows to keep it stable over a long-term period. The share of permissible withdrawal (economic stock) is 23% of its total livestock [17].

The annual costs of reindeer herding on average per 1 hectare of pastures are determined in the data of reindeer farms. They are evaluated on the cost of keeping a standard reindeer herd of 1200 heads, its production and sales [18–20]. The calculations take into account only those costs that are directly related to the decrease in production volumes in reindeer herding. These include: zootechnical processing;

- feeding deer;
- livestock insurance;
- transportation and sales of reindeer products.

Socially significant items such as socio-cultural support, material and technical equipment, salary and training of herders, the work of specialists (veterinarians, zootechnicians, etc.), the construction of production facilities should not be taken into account. Such an approach is due to the fact that traditional activities are not very profitable and are carried out for the purpose of live support in severe environmental conditions. Reducing the cost of reindeer products by the amount of socially significant costs deprives the local population of their means of existence. The share of direct material costs in gross output of reindeer herding is 5.4%. As a result of the calculations, the total annual losses of reindeer herding in the exclusion and stress zones are 2,137,000 rubles ($33,400). The total losses of reindeer husbandry at the estimated site for the entire period of industrial development (5 years) amounted to 10,685,000 rubles ($167,000).

*3.2. Calculation of Losses for Hunting*

Similar to reindeer herding, in the exclusion zone the hunting grounds lose their value completely. In the zone of stress due to the disturbing influence, the productivity of hunting grounds is also reduced by an average of 50%.

Calculation of losses of hunting should be made using the data that contain information on the species and the number of hunting objects permitted for production. However, such materials for most of the northern territories in Russia are missing or outdated. Therefore, it is permissible in the calculations to use hunting quotas established annually in the regions of Russia.

Quotas for hunting on wild animals and the total area of hunting grounds in the region allow us to determine the average value of their number on the researched area. The data on market prices of hunting products are provided by environmental agencies and open sources (internet, publications in the media, analytical reviews, etc.).

On the basis of official data on the economic reserve of hunting animals, indicators of density are calculated per 100 hectares of hunting grounds. Calculations are made for each type of wild animals or birds. It should be noted that official data on the hunting material costs of indigenous peoples are usually not available. In this regard, the costs of hunting are taken as the amount of 5.4% of the

value of gross output. The total annual loss of hunting for the area in the exclusion and stress zones is 242,000 rubles ($3,800). The losses of hunting for the entire period of industrial development (5 years) amounted to 1,210,000 rubles ($19,000).

### 3.3. Calculation of Losses for Fisheries

The Polovinnaya river and its tributaries belong to the Anabar river basin, but not included in the large fishing reservoirs. Fishing is not regular here. According to ichthyologists, the fish productivity of the Polovinnaya is low and amounts to 3 kg of fish per 1 hectare of water surface. The length of the riverbed is 75.4 km, the surface area is 590.5 hectares.

The objects of fishery are pike, perch, grayling, whitefish, Arctic lenok, taimen (hucho) and Artic cod. The indigenous peoples catch the fish mainly for their own food. The fish is sold in small quantities. Price, depending on the type of fish, ranges from 100 to 500 rubles per 1 kg. Since the catch is mainly dominated by relatively low-value (quota-free) species of fish, the average price of 330 rubles/kg is assumed in the calculations.

The overall decrease in allowable catch volumes is equal to the product of the fish productivity index per 1 ha of water surface area over the entire water surface area. For the researched area, the decrease in catch will be 1771.5 kg. The cost of catch loss is equal to the product of the mass of catch reduction by the price of fish. It is 584,595 rubles per year ($1771.5 \times 330$).

The share of costs for fisheries is taken in the size of 5.4% of the value of gross output. The total cost of fishing will be 31,568 rubles ($584,595 \times 0.054$). The average annual size of current losses in the fishing industry is equal to the value of the total losses minus the costs. It is 553 thousand rubles ($8,600).

The losses for the entire period of the partly river exclusion (5 years) amount to 2,765000 rubles ($43,200).

### 3.4. Calculation of Losses for Gathering

The main species of plants in the researched territory include cowberries, blueberries, cloudberries, mushrooms. Other species of wild herbs are rarely met; they practically do not occur in the estimated area and as a result, they are not taken into account during the assessment.

There is a gathering reducing in only the exclusion zone but not in the stress zone. Traditional lands in the exclusion zone are considered as unsuitable for traditional economic activities during the license mine period. In the stress zone, the productivity of wild-growing plants will remain at the same level and therefore losses for this zone are not calculated.

Data on the economic reserves of the plants are determined by geobotanical contours. The cost of the economic reserve for each geobotanical contour is equal to the product of their economic reserve (kg/ha) in the contour by the price of the plant (rubles/kg) and by its area. The summarizing of different reserves gives the total value of economic reserves in the contour. The sum of contours gives the total value of the economic reserves of the plants.

It should be noted that the gathering is a seasonal. It is carried out in a limited period of the year in the most accessible areas around the settlements. Remote territories and areas with low and moderate productivity are practically not used. The gathering is the less productive traditional activity. The total harvest of berries and mushrooms averages 10% of the economic reserves [19]. Gathering is for local consumption only. There are no official data on the gathering costs. Therefore, their share is assumed to be the same as for other types of traditional nature use 5.4% of the value of gross output. The average annual losses due to gathering reducing is equal to the total losses minus the costs. They are 503,000 rubles ($7860). The total losses for gathering for the entire period is 2,515,000 rubles ($39,300).

The total losses (or compensation payments) for the Polovinnaya project is sum of losses for all types of traditional economic activities, Table 1 (all calculations are made by the authors).

**Table 1.** Losses of traditional economic activities in industrial land development.

| Type of Traditional Economic Activity | The Current Value of Losses for the Year (for the Year of Registration of the License, 2017), Rubles/US$ | The Total Amount of Losses for the Entire Period of Development (5 Years), Rubles/US$ |
|---|---|---|
| Reindeer herding | 2,137,000/33,400 | 10,685,000/167,000 |
| Hunting | 242,000/3800 | 1,210,000/19,000 |
| Fishing | 553,000/8600 | 2,765,000/43,200 |
| Gathering | 503,000/7860 | 2,515,000/39,300 |
| Total: | 3,435,000/53,660 | 17,175,000/268,300 |

This case-study shows the methodology of calculation for losses using in some Arctic projects and their cultural impact assessment. During license period the indigenous community will receive the compensation payment of 17,175,000 Rubles ($268,000). The authors suggest to use the methods mentioned above for other mining projects realized on the traditional lands. These methods of losses calculation simplify the process, making it clearer for all Arctic stakeholders.

## 4. Discussion

Despite the measures taken in recent years, the life quality of the indigenous peoples has been complicated by the inability for adaptation to modern economic conditions. The low competitiveness of traditional economic activities is due to low production volumes, high transport costs, the lack of modern enterprises and technologies for the integrated processing of raw materials and biological resources.

The crisis in traditional economic activities has led to the aggravation of social problems. The standard of living of a significant part of citizens living in traditional lands or leading a nomadic lifestyle is lower than the average Russian one. The unemployment rate in the indigenous communities is 1.5–2 times higher than the average in the Russian Federation.

Intensive industrial development of natural resources of the Northern territories has also significantly reduced the possibilities for traditional economic activities [20]. Large areas of reindeer pastures and hunting grounds have been removed from the traditional economy. The environmental problems and rivers' pollution affect traditional crafts, decreasing the fish. The extension of industrial development in the Arctic has revealed deep problems in different spheres of indigenous communities. It is necessary to improve the legislative regulation of the traditional lands, which can become an effective tool for the preservation and development of the traditional way of life and traditional economy. The cultural assessment and benefit sharing are rather new for Russia. They serve as the mechanism for avoiding conflicts in the Arctic and giving the industrial development more civilized characteristics. Unfortunately, the losses of indigenous peoples are inevitably negative consequence. But they must be clearly determined and fairly compensated.

The losses of indigenous peoples are part of the damage to the environment, culture and lifestyle. At present their calculation is an important task for government and science. The problem has been studied by many researchers, but so far it has not been completely resolved. This is due to the lack of accounting in certain types of traditional economic activities [21].

We have developed methodological approaches and an algorithm for losses calculation taking into account the current conditions of economic development. Primarily, these tools are supposed to be used in the traditional lands of the northern territories of Russia. They were tested in seven cultural impact assessments in 2015–2019 in Yakutia. The calculation methodology has positive results. It has allowed indigenous peoples' rights to compensation to be defended, benefit sharing agreements to be concluded, and the subsoil users to participate in social and economic development of the traditional lands. In the Polovinnaya case-study, we have shown the applied side of the methodology.

Our results are the preliminary studies and do not cover all the problems of losses calculation. But they contribute to the important problem-solving: to determine the fair compensation for damage to

the indigenous peoples due to negative mining impact. The issues of compensation and its distribution have yet to be resolved, since this separate important aspect is also not studied in Russia, and there are no legal instruments for this [22].

The authors' approach to losses calculation based on resource assessment have been presented to the Committee for Nationalities of the State Duma in 2019. Nowadays there are hearings on the improvement of this procedure adopted by federal law on the guarantees of indigenous peoples' rights.

As a result of the work on the Polovinnaya project, the methodological approaches developed by the authors were tested for the first time taking into account all factors, impacts and limitations. An algorithm for losses calculation based on the available statistical, regulatory and expert information is presented.

Based on the resource assessment of the territory, the research results made it possible to determine the losses to indigenous peoples living in the license area the project. The amount of compensation payments is justified, and is understandable both for indigenous peoples and subsoil user. Thus conflicts of interest are reduced to a minimum. This mechanism was not previously used in the planned economy of Soviet Union. The proposed approach assumes that there are some limitations that relate to the resource assets of the territory. This is connected with the principle of renewability of the entire biological reserve for each of the types of natural resources (deer pastures, hunting grounds, fish resources, lands for gathering wild berries, herbs). The limitations apply to other less common types of traditional environmental management. Dog breeding, breeding horses and others were not taken into account.

Also, the problem of payments is very important and relevant for modern Russia. There is still no consensus about who is the recipient of compensation payments, and in what form (monetary, non-monetary, etc.): to each resident in the traditional territory; only those who have indigenous status; the tribal community; local government; associations of indigenous peoples (regional branches of the Russian Association of Indigenous Peoples of the North (RAIPON). Currently, this issue is resolved at the legislative level.

It should be also noted that the considered methodology of land valuation has been recently implemented in the practice in Russia and it needs still improvement. In some Arctic regions, there are examples of the interaction between indigenous communities and business. For example, in Yamal, compensation payments are established as a result of negotiations between indigenous peoples and the subsoil user. This leads to the fact that the cost of losses is subjective and does not reflect real indicators. This approach cannot be considered as a scientific. Yakutia is the only region in Russia where the methodology for the land evaluation and its resources is the basis for determining of the compensation. Such evaluation is mandatory for business under regional law. But there are other consumers of this information. The problem is that, due to the lack of financial resources, indigenous peoples and local communities cannot order such research, and therefore do not know the real value of their resources, cannot reasonably defend their economic rights. A limitation of this methodology is connected with the fact that it takes into account only the cost of the biological productivity of the land, but not ecosystem services, the water factor (limits of water access), and climatic aspects. The inclusion of such indicators requires highly qualified specialists, so the issue of education in the field of the evaluation of Arctic resources is also very important. This problem can be solved through international cooperation and knowledge sharing.

In Russia the methodological approach can be used for losses' calculation and compensation payments during other types of mineral deposits' development: gold, hydrocarbons, etc., which will be developed in the future. This methodology can be used both for the northern regions of Russia and other Arctic countries.

**Author Contributions:** Conceptualization, V.G.; Data curation, B.E.B.; Formal analysis, A.A.G.; Funding acquisition, V.G.; Investigation, B.E.B.; Methodology, A.A.G.; Project administration, V.G.; Software, S.I.N.; Supervision, B.E.B.; Visualization, S.I.N.; Writing—original draft, S.I.N.; Writing—review & editing, V.G.

**Funding:** The publication was prepared with the support of the RUDN University Program "5-100"; the Grant of the Russian Foundation for Basic Research (RFBR), project No. 19-010-00023 "Methodology and mechanisms for the distribution of benefits in the industrial development of a territory in the Russian Arctic"; Council on grants of the President of the Russian Federation, Russian scientists - doctors MD-402.2019.6.

**Conflicts of Interest:** The authors declare no conflict of interest.

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
