# Peer review of "Land Resources Evaluation for Damage Compensation to Indigenous Peoples in the Arctic (Case-Study of Anabar Region in Yakutia)"

_resources, doi:10.3390/resources8030143_

Round 1

Reviewer 1 Report

Land assessment in the Arctic as the basis of fair compensation to indigenous peoples for benefits sharing agreements 

 Sergey I. Nosov1, 3, Boris E. Bondarev2, 3, A.A. Gladkov4, Violetta Gassiy

Introduction

The intro has the background but has some grammatical and organization areas.

You are trying to identify ways to support indigenous people from development. Keep that as main theme and keep it simple. The flow in your introduction needs work to help the reader understand and appreciate your data.

Methods

The object of the research is the territories of the traditional economic activities of the indigenous

 peoples, where industrial development is carried out by mining companies.

This is not a good sentence to start your methods. Keep the English simple. For example

The object of this research Is to identify compensation for indigenous peoples for land development.  

The indigenous people in the arctic that are not reindeer herders would argue this statement

The most important activity for the preservation of the traditional lifestyle and culture of these

 peoples is reindeer herding. How about.

One of the most important activity for the preservation of the traditional lifestyle and culture of these

 peoples is reindeer herding.

Very important methodology and would make you paper much stronger

How do you determine habitat quality? Not all habitat is equal. How do you pay for great habitat vs poor habitat?

Have sub section similar to results

Results

Your methods are for reindeer herding but you also talk about other impacts such as hunting.  Much of your results section could be moved to methods section. Simplify the results section and move most of the material to method and discussion.

How can you stop after 5 years won’t the development still be there and impacting herding, etc.

Calculate everything into either dollar or Euro.

Discussion  

You need to discuss why your results turned out the way they did. You also need to discuss impacts of this research. This discussion is very weak and needs improvement.

I like the idea of you paper and inline with a Cost Benefit Analysis, but you need organization, flow, and better explanation. I would recommend getting editing help.

Author Response

Comments and Suggestions for Authors

Land assessment in the Arctic as the basis of fair compensation to indigenous peoples for benefits sharing agreements 

 Sergey I. Nosov1, 3, Boris E. Bondarev2, 3, A.A. Gladkov4, Violetta Gassiy

Introduction

The intro has the background but has some grammatical and organization areas.

You are trying to identify ways to support indigenous people from development. Keep that as main theme and keep it simple. The flow in your introduction needs work to help the reader understand and appreciate your data.

 We decided totally to rewrite the Introduction to make it simpler to understand the relevance of the theme for modern Russia. We try to help reader understand why it is so important and why it is still a problem for scientists, indigenous communities and business. The absence of clear mechanism for losses calculation and traditional lands assessment leads to conflicts and the infringement of the indigenous peoples’ rights. The idea of the paper is to show the improvement of the mechanism offered by the authors.    

Methods

The object of the research is the territories of the traditional economic activities of the indigenous peoples, where industrial development is carried out by mining companies.

This is not a good sentence to start your methods. Keep the English simple. For example

The object of this research Is to identify compensation for indigenous peoples for land development.  Yes, we’ve changed it. Thank you.

The indigenous people in the arctic that are not reindeer herders would argue this statement.

The most important activity for the preservation of the traditional lifestyle and culture of these peoples is reindeer herding. How about.

One of the most important activity for the preservation of the traditional lifestyle and culture of these peoples is reindeer herding. Yes, we’ve change it. Thank you. The authors want to concentrate your attention on the Russian specificity of cultural assessment and damage compensation in the Arctic to emphasize the need to its improvement and further discussion. In Russia the specificity of the indigenous peoples is reindeer herding as the main type of traditional activity comparing Canada, Finland or US. In 2018 the livestock number reached 1.5 million and it still keeps traditional knowledge (language, nomad, equipment). This is crucial to understand the paper and what is the base of the Russian Arctic communities considering in the paper.           

Very important methodology and would make you paper much stronger

How do you determine habitat quality? Not all habitat is equal. How do you pay for great habitat vs poor habitat?

We determine the habitat according the Russian legislation: historically developed area within which indigenous peoples of the North carry out cultural and household activity and which influences their self-identification, a way of life. Thus habitat quality is the degree of provision of the indigenous peoples with material, spiritual and social benefits necessary for keeping their traditional way of life. Economic and social development are parts of the research during cultural assessment. The analysis of the habitat provides the information of possible business investment to the indigenous community according the benefit sharing agreement.

Have sub section similar to results

 We added 2 subsections in Methods:

2.1 The losses calculation and compensation payments as the discussion points in Russian science

2.2 Resources assessment for the losses’ calculation: the way to the fair compensation for traditional lands’ damage

Results

Your methods are for reindeer herding but you also talk about other impacts such as hunting.  Much of your results section could be moved to methods section. Simplify the results section and move most of the material to method and discussion. We are not sure that those part from Research could be moved to methods section. We were trying to show how the methods are processing into results.  We’d like not to move the text. 

How can you stop after 5 years won’t the development still be there and impacting herding, etc. In the paper we said that the losses’ calculation is made for the license period. In this case that period is 5 years. The mining doesn’t start in day. There is period of deposit justification (geological research). If the deposits are confirmed the license could be prolongated for more years. This is term of Russian legislation. The procedure of cultural assessment supposes the monitoring the impact during the exploration period. Then must be another assessment and other calculations for another license period.  

Calculate everything into either dollar or Euro. We have calculating in US$ for better understanding.

Discussion  

You need to discuss why your results turned out the way they did. You also need to discuss impacts of this research. This discussion is very weak and needs improvement. I like the idea of you paper and inline with a Cost Benefit Analysis, but you need organization, flow, and better explanation. I would recommend getting editing help.

We have expanded the Discussion to show more the positive results and to perform further ways of the improvement. We’ve also requested English editing services.

Reviewer 2 Report

This is a worthy and interesting paper.  It needs significant improvement in terms of the English presentation.  But the concepts are clear, the evidence is interesting, and the material is of considerable value to international scholars active in the field.  

I do have some concerns, as follows:

The paper assigns no specific value to cultural losses associated with industrial disruptions;

The social and cultural elements are mentioned in passing; ideally, this would be expanded substantially.  

The paper assigns no value to the strategic significance of the lands under development.  Location is a natural resource, with commercial/economic value.

The paper assumes that Indigenous peoples are compensated only for direct disruptions.  Internationally, this is no longer the case.

The paper does not fully discuss the long-term effects of industrial activity (or, conversely, the fact that some disruptions are short term in nature);

The paper does not address Indigenous priorities and goals.  Globally, they are looking for long-term sustainability and cultural survival and not just for compensation for disruptions.

In sum, the paper does not address many clear and important questions.  This highlights the degree to which this a contribution to Russian as opposed to international scholarship.

I do not think that the paper has to be revised substantially. Rather, the authors should indicate the areas that are not covered substantially in order to highlight their awareness of the limitations of the study.    I would encourage the authors to give some thought to the degree to which Indigenous interest in their territories have economic value that goes well beyond the costs and consequences of industrial displacement.  This is the essence of Indigenous compensation demands and expectations in other countries.  If this is off the table in Russia, it should be mentioned.

Author Response

Comments and Suggestions for Authors

This is a worthy and interesting paper.  It needs significant improvement in terms of the English presentation.  But the concepts are clear, the evidence is interesting, and the material is of considerable value to international scholars active in the field.  

Thank you for reviewing. We’ve requested the English editing services.

I do have some concerns, as follows: The social and cultural elements are mentioned in passing; ideally, this would be expanded substantially.  The paper assigns no specific value to cultural losses associated with industrial disruptions;

In Russia there is still a huge gap in impact assessment tools especially in the context of traditional economy and culture damage. This problem is connected with the absence of cultural assessment legislation which could be used in whole territory of Russia. Nowadays Yakutia is the one region where such norms have been adopted and indigenous communities could consider on fair compensation for damage to traditional lands. They call it ethnological expertise – a comprehensive scientific research on social, economic and cultural impact of the investment project. In the paper the authors describe the case-study of such impact assessment in very narrow term of it. The most problem point of such assessment tool is absence of clear indicators for non-material characteristics of indigenous culture. Thus, to do so the adopted methodology offers to calculate the possible losses of indigenous peoples through their traditional economic activities: reindeer herding, hunting, fishing and gathering. Traditional culture, customs and sacral places must be also considered and described. The idea is that calculation of losses allows to define the compensation amount which could be used for the traditional culture preservation as well as to fix social and economic problems. Obviously, 4 indicators mentioned above are connected with traditional lands as the main assets for indigenous livelihood. That’s why the land assessment is considered as the basis of fair compensation calculation for benefit sharing agreements. The authors want to concentrate your attention on the Russian specificity of cultural assessment and damage compensation in the Arctic to emphasize the need to its improvement and further discussion.         

The paper assigns no value to the strategic significance of the lands under development.  Location is a natural resource, with commercial/economic value.

The paper considers the Arctic traditional lands that are affected by industrial development. The specificity of them are natural resources and their economic value as a part. We are speaking of the dual situation on strategic significance: on one hand the strategic interests of Russia in the Arctic (extraction and other development), on the other – the unique culture and heritage of the indigenous communities that must be preserve. This idea is now reflected in Introduction. 

The paper assumes that Indigenous peoples are compensated only for direct disruptions.  Internationally, this is no longer the case. We agree, but such assessments considered in the paper as well as the benefit sharing system are rather new for Russia. We are only in the way to some shifts on the federal level. The authors want to concentrate your attention on the Russian specificity of cultural assessment and damage compensation in the Arctic to emphasize the need to its improvement and further discussion.

The paper does not fully discuss the long-term effects of industrial activity (or, conversely, the fact that some disruptions are short term in nature); This is not the point of the paper. Generally, we are speaking on the problems of very narrow part of the industrial development, showing the applied side of our work in the Arctic. 

The paper does not address Indigenous priorities and goals.  Globally, they are looking for long-term sustainability and cultural survival and not just for compensation for disruptions. We agree, but here we tried to give more information on this in lines 140-145. But the goal of the paper is more applied character to show how the methodology works.

In sum, the paper does not address many clear and important questions.  This highlights the degree to which this a contribution to Russian as opposed to international scholarship.

I do not think that the paper has to be revised substantially. Rather, the authors should indicate the areas that are not covered substantially in order to highlight their awareness of the limitations of the study.    I would encourage the authors to give some thought to the degree to which Indigenous interest in their territories have economic value that goes well beyond the costs and consequences of industrial displacement.  This is the essence of Indigenous compensation demands and expectations in other countries.  If this is off the table in Russia, it should be mentioned. Yes, it is mentioned in several parts of the paper. Please see the lines 66-89.

Reviewer 3 Report

This paper requires extensive English language translation. If it is to be translated the translator needs to have a good grasp of natural resource management, economics, and Indigenous rights. Google translate, or a generalist translator will not be able to recognize phrases like "ethnological expertise" when I think the authors mean research. The translator needs to know what seasonal rounds are, land use planning, traditional land use studies.

The authors should consider not just the compensation for material loss of land use and food, they should also include spiritual and cultural impacts. When calculating loss, food costs as replacement is one factor, as are the factors related to economic activity (livleihood as well as traditional food security) and the cultural/spiritual connections and impacts.

Cumulative effects should also be part of the inquiry when determining loss. It is not just one immediate project, it is down stream impacts, it is long term and multiple projects and their impacts. Assigning or determining compensation or impacts should also be part of the analysis.

Author Response

Comments and Suggestions for Authors

This paper requires extensive English language translation. If it is to be translated the translator needs to have a good grasp of natural resource management, economics, and Indigenous rights. Google translate, or a generalist translator will not be able to recognize phrases like "ethnological expertise" when I think the authors mean research. The translator needs to know what seasonal rounds are, land use planning, traditional land use studies.

We have significantly improved English but still requested the language editing service.

The authors should consider not just the compensation for material loss of land use and food, they should also include spiritual and cultural impacts. When calculating loss, food costs as replacement is one factor, as are the factors related to economic activity (livleihood as well as traditional food security) and the cultural/spiritual connections and impacts.

In Russia there is still a huge gap in impact assessment tools especially in the context of traditional economy and culture damage. This problem is connected with the absence of cultural assessment legislation which could be used in whole territory of Russia. Nowadays Yakutia is the one region where such norms have been adopted and indigenous communities could consider on fair compensation for damage to traditional lands. They call it ethnological expertise – a comprehensive scientific research on social, economic and cultural impact of the investment project. In the paper the authors describe the case-study of such impact assessment in very narrow term of it. The most problem point of such assessment tool is absence of clear indicators for non-material characteristics of indigenous culture. Thus, to do so the adopted methodology offers to calculate the possible losses of indigenous peoples through their traditional economic activities: reindeer herding, hunting, fishing and gathering. Traditional culture, customs and sacral places must be also considered and described. The idea is that calculation of losses allows to define the compensation amount which could be used for the traditional culture preservation as well as to fix social and economic problems. Obviously, 4 indicators mentioned above are connected with traditional lands as the main assets for indigenous livelihood. That’s why the land assessment is considered as the basis of fair compensation calculation for benefit sharing agreements. The authors want to concentrate your attention on the Russian specificity of cultural assessment and damage compensation in the Arctic to emphasize the need to its improvement and further discussion.         

Cumulative effects should also be part of the inquiry when determining loss. It is not just one immediate project, it is down stream impacts, it is long term and multiple projects and their impacts. Assigning or determining compensation or impacts should also be part of the analysis. We absolutely agree with your comment but this is not the goal of the paper. Generally, we are speaking on the problems of very narrow part of the industrial development, showing the applied side of our work in the Arctic.  Land assessments, calculation of losses considered in the paper as well as the benefit sharing system are rather new for Russia. We have started to work with such definitions in applied researches only in 2000-ths in Russia. Hope, there is no need to explain that in Soviet Union or later in 1990-ths the benefit sharing issues didn’t exist in Russia and never has been mentioned. That’s why the applied research on that narrow part of the benefit sharing agreement is need to be performed now.       

Round 2

Reviewer 1 Report

There are still some basic grammatical mistakes that can be easily fixed with a quick round of edits I have given you some examples below.

A few citations  needed on between line 56 and 71

Line 75

If in Canada or US such systems have been successfully working and economic rights of indigenous peoples are clear defended.

What are you trying to say here? clarify

Here we can remember the Makivik Corporation and its highly developed system of the business collaboration.

Explain and clarify

Line 81 add the in front of whole

Line 89 sacred

I think the paper reads much better and the sections are much clearer. I really like the case study style for the results

I would still like to see a bit more in the discussion

1st  what did you find in the case studies that was unknown or qualified by your data,

2nd what are larger ramifications        

3rd limitations,

4th how can you methods be used to the upmost

5th can it only be used in Russia or can it be expanded

Author Response

Dear Reviewer,

Thank you for your comments. We tried to follow your recommendations.

1. We have changed the title and abstract of the paper (to make it more applied to the researched area):

Land resources evaluation for damage compensation to indigenous peoples in the Arctic (case-study of Anabar region in Yakutia)

Abstract: The compensation for losses caused to the indigenous peoples in Arctic Russia due to the industrial development of their traditional lands is the most urgent question which resolution requires development of new mechanisms and tools. The losses caused to indigenous traditional lands are part of the damage caused to the natural environment, their culture and livelihood. In the Russian Federation the cultural impact assessment is rather new tool aiming to protect indigenous peoples’ rights to lands. In this paper the authors show the applied side of the cultural assessment which is used to improve the methodology of calculation of losses adopted by Ministry of regional development in Russia in 2009. This methodology is based on the resource disposition and evaluation of the traditional lands. Accordingly, compensation payments are calculated as the sum of the losses in traditional economic activities such as: reindeer herding, hunting, fishing and gathering. Such compensations are considered by authors as the elements of the benefit sharing system. In practice, this methodology has been tested at industrial projects on alluvial diamonds in Yakutia. In this paper we look at the Polovinnya project case-study which deals with Indigenous peoples of Dolgans and Evenks and argue that such methodology developed for indigenous peoples and subsoil user could reduce to a minimum the conflict of interests.

2. There are still some basic grammatical mistakes that can be easily fixed with a quick round of edits I have given you some examples below.

A few citations  needed on between line 56 and 71. We added the citation [3].

Line 75. If in Canada or US such systems have been successfully working and economic rights of indigenous peoples are clear defended. What are you trying to say here? clarify

3. We added in the text (line 74-93)

In these countries there is a clear division of territories according to the principles of ownership, the status of indigenous communities and their rights to traditional lands. In Russia there is still the gap in the legislation of the tribal families’ ownership of the traditional lands. At present, the issue of the conditions for the land provision to indigenous peoples for the maintenance of traditional lifestyles and management is regulated by the legislative acts ambiguously and incompletely. It would seem that special legislation establishes the right of minority peoples to the free use of land in order to protect their traditional way of life, business and crafts. It is contrary to other legislation, where restrictions are provided. Thus, according to Article 10 of the Federal Law No. 101-FZ “On the turnover of agricultural land”, land plots from agricultural land that are state or municipal property may be leased to indigenous communities. At the same time, they also cannot buy out the leased land in the property. The same situation is with hunting. Today, the hunting grounds are put up for auction according to the federal legislation. If the indigenous hunter is without stable work today how he could buy out these lands? Fishing is the main form of survival and nutrition in the Arctic and other Northern regions. The transition of the fishery to auction also outraged the population. These are the conflicts where the law on the one hand protects indigenous peoples, and on the other the Forest Code infringes upon their right to traditional and ancestral lands. The indigenous peoples living in territories that have received the status of specially protected areas should not be restricted to the extraction of hunting resources for personal consumption. Today, northerners feel uncomfortable and cannot freely hunt in the tundra or fish, waiting for fines each time. The traditional occupations of the indigenous peoples of the North — reindeer herding, fishing, hunting, are closely interrelated with the survival of these peoples in their native land.

Here we can remember the Makivik Corporation and its highly developed system of the business collaboration. Explain and clarify

4. Line 94-104:

In the Canadian province of Quebec, the Makivik Corporation (“The Makivik Corporation”) is a significant economic force, an ethnic corporation whose investment interests are represented in areas such as oil and gas production, transportation, environmental protection, etc. Owning equity stakes in these areas over thirty years The corporation invested more than $ 100 million in social infrastructure and community support projects. In 2002, signed a trilateral agreement for 25 years on partnership in the field of socio-economic development of Nunavik (Nunavik) territory joined in 1999 in Canada - a place of compact residence for the indigenous people of the Inuit (Eskimos). The parties to the signing were the Government of Quebec, the regional authorities (The Kativik Regional Government) and the Makivik Corporation. According to this agreement, the main objects of joint investment are mining industry, tourism, transport and social infrastructure, construction of hydroelectric power plants, environmental protection.

I think the paper reads much better and the sections are much clearer. I really like the case study style for the results

I would still like to see a bit more in the discussion

1st  what did you find in the case studies that was unknown or qualified by your data,

2nd what are larger ramifications        

3rd limitations,

4th how can you methods be used to the upmost

5th can it only be used in Russia or can it be expanded

5. We added to the Discussion (line 444-460):

As a result of the work on the Polovinnaya project, the methodological approaches developed by the authors were tested for the first time taking into account all factors, impacts and limitations. An algorithm for losses calculating based on the available statistical, regulatory and expert information is presented.

Based on the resource assessment of the territory, the research results made it possible to determine the losses to indigenous peoples living in license area the project. The amount of compensation payments is justified, understandable both for indigenous peoples and subsoil user. Thus conflict of interest is reduced to a minimum. This mechanism was not previously used in the planned economy of Soviet Union. The proposed approach assumes that there are some limitations that relate to the resource assets of the territory. This is connected with the principle of renewability of the entire biological reserve for each of the types of natural resources (deer pastures, hunting grounds, fish resources, lands for gathering wild berries, herbs). The limitations apply to other less common types of traditional environmental management. Dog breeding, breeding horses and others were not taken into account. These methodological approaches can be used for losses’ calculation and compensation payments during other types of mineral deposits’ development: gold, hydrocarbons, etc., which will be developed in the future. This methodology can be used both for the northern regions of Russia and other Arctic countries.

5. Also we have requested English editing services.

Reviewer 3 Report

Title suggestion: "Land assessment" in the Russian Arctic as a tool for fair compensation in benefit sharing agreements with Indigenous Peoples

Otherwise the title now is too general and certainly isn't comparative with other circumpolar nations and other Indigenous peoples.

I would rather the specific Indigenous peoples in question or who have seen "fair compensation" are actually named. If for example it is RAIPON or a specific peoples/nation then name them, each time, rather than apply generic terms.

The English translation/edit is still not sufficient for an academic peer reviewed journal. I mean no disrespect and am not commenting on the original Russian.

First sentence of the abstract, as an example. I think the authors mean:

In the Russian Arctic Indigenous land use and "ownership" rights predate the formation of the Russian state. These rights to land and use of traditional resources are recognized by the United Nations Declaration for the Rights of Indigenous Peoples (UNDRIP). Industrial development in the Russian Arctic, for example mining and oil and gas extraction, significantly impact Indigenous rights, life-ways and economies, so much so that the compensation for loss is an important consideration for the state and corporations.  

The authors should consider this a narrative, we will argue X (your thesis) and here is the evidence (proof for it, maybe also the counter arguments and why they are dismissed), followed by the conclusion. Keep it simple and straight forward, don't hide behind jargon.

Author Response

Dear Reviewer,

Thank you for your comments. We tried to follow your recommendations.

Title suggestion: "Land assessment" in the Russian Arctic as a tool for fair compensation in benefit sharing agreements with Indigenous Peoples. Otherwise the title now is too general and certainly isn't comparative with other circumpolar nations and other Indigenous peoples. I would rather the specific Indigenous peoples in question or who have seen "fair compensation" are actually named. If for example it is RAIPON or a specific peoples/nation then name them, each time, rather than apply generic terms. The English translation/edit is still not sufficient for an academic peer reviewed journal. I mean no disrespect and am not commenting on the original Russian. First sentence of the abstract, as an example. I think the authors mean:

In the Russian Arctic Indigenous land use and "ownership" rights predate the formation of the Russian state. These rights to land and use of traditional resources are recognized by the United Nations Declaration for the Rights of Indigenous Peoples (UNDRIP). Industrial development in the Russian Arctic, for example mining and oil and gas extraction, significantly impact Indigenous rights, life-ways and economies, so much so that the compensation for loss is an important consideration for the state and corporations.  

The authors should consider this a narrative, we will argue X (your thesis) and here is the evidence (proof for it, maybe also the counter arguments and why they are dismissed), followed by the conclusion. Keep it simple and straight forward, don't hide behind jargon.

1. We have changed the title and abstract of the paper:

Land resources evaluation for damage compensation to indigenous peoples in the Arctic (case-study of Anabar region in Yakutia)

Abstract: The compensation for losses caused to the indigenous peoples in Arctic Russia due to the industrial development of their traditional lands is the most urgent question which resolution requires development of new mechanisms and tools. The losses caused to indigenous traditional lands are part of the damage caused to the natural environment, their culture and livelihood. In the Russian Federation the cultural impact assessment is rather new tool aiming to protect indigenous peoples’ rights to lands. In this paper the authors show the applied side of the cultural assessment which is used to improve the methodology of calculation of losses adopted by Ministry of regional development in Russia in 2009. This methodology is based on the resource disposition and evaluation of the traditional lands. Accordingly, compensation payments are calculated as the sum of the losses in traditional economic activities such as: reindeer herding, hunting, fishing and gathering. Such compensations are considered by authors as the elements of the benefit sharing system. In practice, this methodology has been tested at industrial projects on alluvial diamonds in Yakutia. In this paper we look at the Polovinnya project case-study which deals with Indigenous peoples of Dolgans and Evenks and argue that such justified, understandable methodology both for indigenous peoples and subsoil user could reduce to a minimum the conflict of interests.

2. We added in the text (line 74-93)

In these countries there is a clear division of territories according to the principles of ownership, the status of indigenous communities and their rights to traditional lands. In Russia there is still the gap in the legislation of the tribal families’ ownership of the traditional lands. At present, the issue of the conditions for the land provision to indigenous peoples for the maintenance of traditional lifestyles and management is regulated by the legislative acts ambiguously and incompletely. It would seem that special legislation establishes the right of minority peoples to the free use of land in order to protect their traditional way of life, business and crafts. It is contrary to other legislation, where restrictions are provided. Thus, according to Article 10 of the Federal Law No. 101-FZ “On the turnover of agricultural land”, land plots from agricultural land that are state or municipal property may be leased to indigenous communities. At the same time, they also cannot buy out the leased land in the property. The same situation is with hunting. Today, the hunting grounds are put up for auction according to the federal legislation. If the indigenous hunter is without stable work today how he could buy out these lands? Fishing is the main form of survival and nutrition in the Arctic and other Northern regions. The transition of the fishery to auction also outraged the population. These are the conflicts where the law on the one hand protects indigenous peoples, and on the other the Forest Code infringes upon their right to traditional and ancestral lands. The indigenous peoples living in territories that have received the status of specially protected areas should not be restricted to the extraction of hunting resources for personal consumption. Today, northerners feel uncomfortable and cannot freely hunt in the tundra or fish, waiting for fines each time. The traditional occupations of the indigenous peoples of the North — reindeer herding, fishing, hunting, are closely interrelated with the survival of these peoples in their native land.

Line 95-107:

In the Canadian province of Quebec, the Makivik Corporation (“The Makivik Corporation”) is a significant economic force, an ethnic corporation whose investment interests are represented in areas such as oil and gas production, transportation, environmental protection, etc. Owning equity stakes in these areas over thirty years The corporation invested more than $ 100 million in social infrastructure and community support projects. In 2002, signed a trilateral agreement for 25 years on partnership in the field of socio-economic development of Nunavik (Nunavik) territory joined in 1999 in Canada - a place of compact residence for the indigenous people of the Inuit (Eskimos). The parties to the signing were the Government of Quebec, the regional authorities (The Kativik Regional Government) and the Makivik Corporation. According to this agreement, the main objects of joint investment are mining industry, tourism, transport and social infrastructure, construction of hydroelectric power plants, environmental protection.

3. We added to the Discussion (line 446-467):

As a result of the work on the Polovinnaya project, the methodological approaches developed by the authors were tested for the first time taking into account all factors, impacts and limitations. An algorithm for losses calculating based on the available statistical, regulatory and expert information is presented. Based on the resource assessment of the territory, the research results made it possible to determine the losses to indigenous peoples living in license area the project. The amount of compensation payments is justified, understandable both for indigenous peoples and subsoil user. Thus conflict of interest is reduced to a minimum. This mechanism was not previously used in the planned economy of Soviet Union. The proposed approach assumes that there are some limitations that relate to the resource assets of the territory. This is connected with the principle of renewability of the entire biological reserve for each of the types of natural resources (deer pastures, hunting grounds, fish resources, lands for gathering wild berries, herbs). The limitations apply to other less common types of traditional environmental management. Dog breeding, breeding horses and others were not taken into account. Also the problem of payments is very important and relevant for modern Russia. There is still no consensus about who is the recipient of compensation payments, and in what form (monetary, non-monetary, etc.): to each resident in the traditional territory; only those who has indigenous status; tribal community; local government; associations of indigenous peoples (regional branches of RAIPON). Currently, this issue is resolved at the legislative level. These methodological approaches can be used for losses’ calculation and compensation payments during other types of mineral deposits’ development: gold, hydrocarbons, etc., which will be developed in the future. This methodology can be used both for the northern regions of Russia and other Arctic countries.

4. Also we have requested English editing services

This manuscript is a resubmission of an earlier submission. The following is a list of the peer review reports and author responses from that submission.

Round 1

Reviewer 1 Report

The overall concept is a worthwhile one that requires research to support policy development on benefit sharing to indigenous peoples in the Arctic region under the jurisdiction of the Russian government.

Yet the paper is poorly structured and badly written. I do think that the ideas are within it and the evidence to support those is within the research but the quality fo the argument being made is poor. You do not use the evidence well nor present it in a coherent manner to support the claims made.

There is also limited questioning of the Russian government's actual policies and whether there is substantive commitment to the sustainable development of the Arctic region.

Reviewer 2 Report

The paper is so poorly written it was very hard to follow. The idea of the paper was great, and I was excited to read it. The concept is there but the structure and writing of the paper needs a complete rewrite. I stopped going line by line because it would take days to edit this paper correctly. I also do not trust the methods because it was described so poorly I was not sure I was following the methods correctly.

Line 32. Remove currently

Line 35 remove the

Line 37 – 44 indent the list

Line 41 remove the main

List c sentence is very awkward.

Line 46-47 awkward sentence

Line 50-51 awkward sentence Also the species part of the sentence is a different concept.

Line 55 dog breeding then you mention reindeer – just change to breeding

Line 52 – 65 this list needs to be redone maybe into a table. It is very hard to read

Introduction is very weak and needs a lot more background. I also did not get a good idea in the introduction what you are doing. Needs way more detail.

methods

Line 72-73 awkward sentence

In method make a subsection of study area and clearly describe the region. Please use a better map – even google earth would be a better map than the one used.

Lines 100 – 102 is very good and you need to introduce the concept in the introduction,

Figure 2 what does rub mean in the figure.

What is the algorithm. I want the formula in the methods. How did you value the various parts of the traditional lifestyle?

Need way more methods to explain your study. There is no way someone could read you methods and repeat your design.

Results.

Much of the results should be in methods.

The results section is written very poorly.

Discussion

What is this. I need a great deal more explanation what was found. What are the impacts, how could it be used

Reviewer 3 Report

The paper is well written, interesting, and easy to read.

I didn’t see any major problems in the writing.  The methodology is good and the topic is very relevant to Agricultural Economics.

The only weakness is the final discussion. The final discussion could  be expanded.

In line 45, does the reference list apply only to line 45? Or to lines 35-45. I am assuming is applies to 35-45. The reference is definitely needed for lines 35-45. I found it at The International Expert Council on Cooperation in the Arctic.

Reviewer 4 Report

-       Some sections are poorly developed and unclear

-       Whilst the analysis has been robustly designed and implemented, the
critical discussion of its results and their implications is not advanced to
standard expected for a scientific journal.

- The introduction section of the paper suitably does not motivate the readers in the subject of the paper.

- The paper also lacks a proper discussion section, including limitations and comparison with other papers. 

- Conclusions section should be developed to highlight the unique contributions of the paper, limitations of the research and some future research directions.